# Green Synthesis of NiO Nanoflakes Using Bitter Gourd Peel, and Their Electrochemical Urea Sensing Application

**DOI:** 10.3390/mi14030677

**Published:** 2023-03-19

**Authors:** Irum Naz, Aneela Tahira, Aqeel Ahmed Shah, Muhammad Ali Bhatti, Ihsan Ali Mahar, Mehnaz Parveen Markhand, Ghulam Murtaza Mastoi, Ayman Nafady, Shymaa S. Medany, Elmuez A. Dawi, Lama M. Saleem, Brigitte Vigolo, Zafar Hussain Ibupoto

**Affiliations:** 1Dr. M.A Kazi Institute of Chemistry, University of Sindh, Jamshoro 76080, Pakistan; nazirum11@gmail.com (I.N.); aneela.tahira@salu.edu.pk (A.T.); gm.mastoi@usindh.edu.pk (G.M.M.); 2Institute of Chemistry, Shah Abdul Latif University, Khairpur Mirs 66111, Pakistan; maanomarkhand232@gmail.com; 3Wet Chemistry Laboratory, Department of Metallurgical Engineering, NED University of Engineering and Technology, University Road, Karachi 75270, Pakistan; aqeelshah@cloud.neduet.edu.pk; 4Centre for Environmental Sciences, University of Sindh, Jamshoro 76080, Pakistan; 5Department of Chemistry, College of Science, King Saud University, Riyadh 11451, Saudi Arabia; anafady@ksu.edu.sa; 6Department of Chemistry, Faculty of Science, Cairo University, Cairo 12613, Egypt; shymaasamir80@yahoo.com; 7Nonlinear Dynamics Research Centre (NDRC), Ajman University, Ajman P.O. Box 346, United Arab Emirates; 8Biomolecular Science, Earth and Life Science, Amsterdam University, De Boelelaan 1 105, 1081 HV Amsterdam, The Netherlands; lama.saleem@vu.nl; 9Institut Jean Lamour, CNRS-Université de Lorraine, F-54000 Nancy, France; brigitte.vigolo@univ-lorraine.fr

**Keywords:** bitter gourd peel extract, NiO nanostructures, urea oxidation, non-enzymatic sensor

## Abstract

To determine urea accurately in clinical samples, food samples, dairy products, and agricultural samples, a new analytical method is required, and non-enzymatic methods are preferred due to their low cost and ease of use. In this study, bitter gourd peel biomass waste is utilized to modify and structurally transform nickel oxide (NiO) nanostructures during the low-temperature aqueous chemical growth method. As a result of the high concentration of phytochemicals, the surface was highly sensitive to urea oxidation under alkaline conditions of 0.1 M NaOH. We investigated the structure and shape of NiO nanostructures using powder X-ray diffraction (XRD) and scanning electron microscopy (SEM). In spite of their flake-like morphology and excellent crystal quality, NiO nanostructures exhibited cubic phases. An investigation of the effects of bitter gourd juice demonstrated that a large volume of juice produced thin flakes measuring 100 to 200 nanometers in diameter. We are able to detect urea concentrations between 1–9 mM with a detection limit of 0.02 mM using our urea sensor. Additionally, the stability, reproducibility, repeatability, and selectivity of the sensor were examined. A variety of real samples, including milk, blood, urine, wheat flour, and curd, were used to test the non-enzymatic urea sensors. These real samples demonstrated the potential of the electrode device for measuring urea in a routine manner. It is noteworthy that bitter gourd contains phytochemicals that are capable of altering surfaces and activating catalytic reactions. In this way, new materials can be developed for a wide range of applications, including biomedicine, energy production, and environmental protection.

## 1. Introduction

Urea is a major component of nitrogen sources used in fertilizers and chemical industry applications [1]. Various industrial products contain urea, including soaps, detergents, cleaning agents, and supplementary feed for animals [2]. As a result of its intensive use as a spray agent on crops, urea can easily pollute the environment, causing harm to both life and the environment. The metabolism of proteins can lead to the production of urea in biological systems [3]. Approximately 2.6 to 6.5 mM of urea molecules are dissolved in the fluids of our bodies [4,5], whereas 490 to 2690 mL of urea are dissolved in human urine [6]. Due to the fact that urea is one of the major adulterants in milk, the dairy industry requires highly sensitive and selective methods of detecting urea. Proteins, minerals, and vitamins are among the many nutrients found in milk. According to statistics, milk contains an average of 3.4% protein. Generally, milk contains between 3.1 and 6.6 mM urea, but Indian food safety standards define urea levels at 11.6 mM [7]. To increase the nitrogen content of diluted milk, an additional amount of urea is added [8,9]. Therefore, it is imperative that the exact amount of urea adulteration in milk be regulated in order to prevent the harmful effects of urea on humans. Furthermore, high urea levels in the body fluids can result in kidney failure, urinary tract infections, and obstructions, whereas low urea levels are associated with liver dysfunction, renal dysfunction, and cachexia [10]. There is therefore a need to determine urea with precision and accuracy in a wide variety of applications, including agriculture, biological fluids, food, pharmaceuticals, and environmental regulation [11]. Consequently, a number of analytical methods have been developed to detect urea, including near-infrared spectroscopy [12], chromatography [13], nuclear magnetic resonance [5], flow injection [14], and electrochemical techniques [15,16,17,18]. Recently, electrochemical and electroanalytical techniques have been extensively studied due to their simplicity, low limit of detection, on-site analysis, low cost, rapidity, sensitivity, and selectivity [19]. An enzymatic [20] and a non-enzymatic [21,22] electrochemical approach has been used to quantify urea. The enzymatic urea biosensor has certain limitations, such as the process of enzyme immobilization, and the stability issues associated with the urease enzyme under experimental conditions such as pH, temperature, and humidity [7]. On the other hand, non-enzymatic urea sensing is performed by oxidation/reduction of appropriately modified electrodes [23]. To develop an electrochemical urea sensor that is highly efficient and selective without the use of enzymes, new and surface-tunable materials are required. Consequently, a variety of materials have been investigated for application to the development of electrochemical urea sensors, such as noble and nonprecious metals, metal oxides, and metal hydroxides [24,25,26,27,28,29]. As a result of their tunable surface properties, significant stability under electrolytic conditions, and the potential of growing many nanostructured morphologies, metal oxides are being extensively researched as sensing materials [15,17,18,21]. Among those commonly used to detect urea are nickel-based catalysts due to their less hazardous effects, their cost effectiveness, and their large tunable catalytic sites [30]. K. Boggs et al. [31] have investigated the direct electrochemical urea oxidation mechanism on nickel-based materials by establishing a large number of intermediate species of nickel oxyhydroxide (NiOOH). Although nickel-based materials have been developed extensively for the determination of urea, non-enzymatic methods are still capable of producing wide linear ranges and low defection limits. The chemical routes used for the synthesis of nanostructured materials involve costly chemicals and physical methodologies that are associated with high levels of toxicity, cytotoxicity, and carcinogenicity [32]. However, green methods have been utilized to produce nanostructured materials with enhanced functionality, low cost, and limited toxic effects [33,34,35,36,37,38]. Thus, green chemical synthesis of nanostructured materials is referred to as an environmentally friendly and eco-friendly process. A variety of favorable components are included in the plant biomass waste of plant extract, including capping agents, stabilizing agents, and reducing agents, which play a crucial role in preparing nanostructured compounds that are well controlled in terms of morphology, size, and surface properties [39,40,41,42,43,44]. 

Since NiO nanostructures were green synthesized for nonenzymatic urea determination, no studies have been conducted to assess their catalytic effect, their dimensions, and their shapes, or the nonenzymatic determination of urea, particularly for large samples and real samples, where the matrix effect should be considered as a significant indicator. In our research on practical non-enzymatic urea sensors made of NiO materials, as well as a green chemical approach to NiO nanostructure synthesis, we examined various phytochemicals found in bitter gourd peel extract that are capable of acting as reduction and capping agents for surface modification [45,46,47,48]. Recently, various NiO-based urea sensors have been reported [22,49,50,51]. However, there is still room to fabricate highly efficient enzyme-free urea sensors. Generally, the bitter gourd extract contains a variety of natural reducing agents, capping agents, and stabilizing agents such as proteins, carbohydrates, amino acids, strolls, terpenoids, flavonoids, cardiac glycosides, etc. These various reducing agents and stabilizing agents can provide well-controlled NiO nanostructures with significant surface modifications that are highly desirable for electrocatalytic reactions. In the absence of studies, bitter gourd peel extract has not been used to modify the surface of NiO nanostructures or to develop non-enzymatic urea sensors for various real-world applications. Our study investigated the phytochemical synthesis of NiO nanostructures from bitter gourd peel extract and their application to catalytic urea oxidation under alkaline conditions.

## 2. Experimental Section 

### 2.1. Chemicals Used

All chemicals were purchased from Sigma-Aldrich Karachi, Sindh Pakistan. These samples were all of high analytical quality and were used as received. Nickel chloride hexahydrate, sodium hydroxide, ammonia (33%), urea, Nafion (5%), magnesium chloride, ascorbic acid, glucose, and uric acid. Prior to the experiment, the glassware was cleaned and rinsed with deionized water. All of the glassware was dried in an oven at 60 °C for 30 min. A 0.1 M sodium hydroxide solution was used as a supporting electrolyte. All the solutions for electrochemical measurements were completed in aqueous 0.1 M sodium hydroxide. Deionized water was used as a solvent.

### 2.2. Phytochemical Synthesis of NiO Nanostructures Using Low-Temperature Aqueous Chemical Growth Method

NiO nanostructures were prepared by low-temperature aqueous chemical growth method followed by thermal annealing in the air. A pristine NiO sample was prepared by dissolving nickel chloride hexahydrate (0.1 M) in 200 mL of deionized water. Following this, 10 mL of an aqueous solution of ammonia at a concentration of 33% was added. In addition, 10 and 15 mL of bitter gourd peel extract were added separately to two other beakers containing nickel chloride hexahydrate and aqueous ammonia. There were two samples, referred to as sample 1 and sample 2. During low-temperature aqueous chemical growth method at 95 °C for five hours, growth solutions were tightly covered with aluminum sheets. It resulted in the formation of a light green nickel hydroxide product on the filter paper, after which it was washed several times with deionized water and allowed to dry overnight. Air calcination at 500 °C was then carried out for five hours. Similarly, pristine NiO was synthesized without the use of bitter gourd peel extract. A schematic illustration of the phytochemical synthesis is provided in Figure 1.

### 2.3. Structural Characterization of NiO Nanostructures

For analyzing the crystal quality of NiO nanostructures, powder X-ray diffraction was used at 45 kV, 45 mA, with X-rays emitted from a Cu anode as the source of diffraction patterns. The X-rays were scanned at an angle of 10–85° two-theta. We investigated the shape and orientation of nanostructured NiO, by depositing a few amounts of NiO sample on the conducting carbon tape, using low-resolution scanning electron microscopy at an accelerating potential of 20 kV.

### 2.4. Electrochemical Oxidation of Urea in Alkaline Media Using Modified Glassy Carbon Electrode (GCE)

We prepared a suspension of NiO nanostructures by dispersing 10 mg of each sample in 3 mL of deionized water. In a subsequent step, 30 μL of 5% Nafion was poured into NiO suspension in an ultrasonic bath and stirred for 20 min. Prior to the modification of GCE, the surface was cleaned with silicon paper and alumina paste, followed by washing with deionized water and drying at room temperature with air. A suspension of 10 µL with a loading mass of (0.2 mg) was then applied to the surface of GCE and dried at 50 °C in an electric oven for five minutes. In the electrochemical cell system, NiO-modified GCE served as the working electrode, silver–silver chloride (Ag/AgCl) filled with (3.0 M KCl) served as the reference electrode, and platinum wire served as the counter electrode. The working of GCE was exhibiting an area of 3 mm and it was used to deposit the catalytic material. The urea stock solution was prepared using a solution of 0.1 M NaOH. Various electrochemical methods were used to analyze the results, including cyclic voltammetry, electrochemical impedance spectroscopy, and chronoamperometry. As part of the scan rate study, CVs were measured at various scan rates. The purpose of this study was to examine electrochemical impedance spectroscopy (EIS) at 100 kHz to 0.1 Hz, a sinusoidal potential of 10 mV, and a biasing potential of 0.5 V. The raw EIS data were simulated using Z-view software, and well-fitting results were obtained. In order to determine the interference pattern and stability of the modified electrode, chronoamperometry was used. We conducted all measurements at room temperature. In order to determine the urea concentration in the real samples, 1:10 dilutions of the real samples into 0.1 M NaOH were made before the spiking for direct measurement of urea in curd, milk, wheat flour, blood, and urine. Blood and urine samples were collected from human subjects represented by laboratory personnel with their consent for use in our research.

## 3. Results and Discussion 

### 3.1. The Structural Characterization of NiO Nanostructures Prepared with Bitter Gourd Peel Extract 

As shown in Figure 1a,b, powder XRD was used to examine the crystal quality of NiO nanostructures prepared with bitter gourd. The diffraction patterns of NiO were primarily characterized by the cubic phase, which is supported by the standard JCPDS card no. 01-073-1523. A number of reflection peaks have been identified, which correspond to typical crystal planes in NiO, including 111, 200, 220, 311, and 222. The intense peak observed in the direction patterns at the crystal planes 111 and 200 are well matched with the reported work [52], and the reference (JCPDS card No. 00-001-1258). It was confirmed that all bitter gourd NiO samples were free of other impurities. Phytochemicals in bitter gourd juice have induced two theta shifts to higher angles on NiO crystals, as shown in Figure 1b. The shift in angle may be attributed to the expansion of the crystal size caused by the biopolymers in bitter gourd juice. This could have resulted in defects in the prepared material. An image of the morphology of NiO nanostructures synthesized with bitter gourd is shown in Figure 1. The pristine NiO sample is depicted in Figure 1c as having a sheet-like morphology. As shown in Figure 1d, NiO nanostructures prepared using 10 mL bitter gourd peel extract also showed thin NiO flakes oriented as flowers. Figure 1e shows that NiO nanostructures exhibited a floral appearance after being treated with 15 mL of bitter gourd peel extract. The bitter gourd contains a variety of phytochemicals, such as reducing agents and capping agents, which together shape the surface features of NiO nanostructures. The thickness of the NiO flakes with flower-like morphology could range from 100 to 200 nanometers, indicating remarkable nanoscale features, and such dimensions are associated with high surface areas, resulting in efficient catalytic performance.

### 3.2. Non-Enzymatic Urea Sensor Characterization Based on the Bitter Gourd Peel Extract Assisted NiO Nanostructures

As shown in Figure 2, preliminary electrochemical characterization was conducted using cyclic voltammetry in alkaline conditions with 0.1 M NaOH on the bare glassy carbon electrode (GCE) and the modified glassy carbon electrode. Two samples of NiO were obtained with 10 and 15 mL of biomass waste from a bitter gourd by modifying the GCE with pristine NiO nanostructures. As can be seen, GCE did not possess any redox properties at its simplest level. Despite this, NiO samples have been found to exhibit redox characteristics under alkaline conditions of 0.1 M NaOH. As part of the development of non-enzymatic sensors, NiO nanostructures were tested on a wide range of biomolecules, including glucose, uric acid, ascorbic acid, dopamine, and urea. However, NiO nanostructures were found to be active only in the nonenzymatic oxidation of urea. As a result, we are mainly reporting non-enzymatic characterizations of the urea biomolecule. The addition of 0.1 mM of urea significantly enhanced the redox properties of sample 1 of NiO nanostructures prepared from bitter gourd peel extract. Figure 2b illustrates this concept. Figure 2b illustrates how the phytochemicals present in 10 mL of bitter gourd have significantly modified the surface properties of NiO nanostructures.

The catalytic activity of NiO nanostructures was further decreased by the addition of bitter gourd. The adverse effects are due to the presence of a high concentration of phytochemicals in 15 mL of bitter gourd juice. The increased oxidation peak current in sample 1 indicates better electron transfer in the recently designed architecture with tailored surface properties. In Figure 2a, CV curves are shown for the different materials in the presence of an electrolyte of urea in order to understand their electrochemical performance. Different materials exhibit apparent redox behavior when electrochemically treated in an electrolytic solution. Furthermore, it was estimated that the peak current for sample 1 in 0.1 M NaOH was approximately 1.08 µA, whereas the peak current for sample 1 in 0.1 M urea was approximately 1.27 µA. A difference of around 0.19 µA was observed between sample 1 in 0.1 M NaOH and sample 1 in the absence of urea, indicating sample 1 has significant catalytic properties for the oxidation of urea. A comparison of CV curves of different materials in the presence of 0.1 mM urea is shown in Figure 2b. The peak current enhancement for sample 1 in the presence of urea is higher than the peak current for the pristine sample, suggesting that sample 1 has experienced favorable urea oxidation on its surface. Under alkaline conditions, the sensing mechanism of urea on the surface of NiO nanostructures can be illustrated as shown below. Based on a forward scan of the CV curve (Equation (1)), the applied potential results in the oxidation of Ni(II) species present on the nanostructured surface of NiO to Ni(III). The reverse scanning of CV (Equation (2)) [53,54,55,56,57,58] shows that Ni(III) species are highly active towards urea oxidation and then become Ni(II) species as a result of electron transfer.
Ni(OH)_2_ + OH^-^ → NiOOH + H_2_O+ e^-^(1)
CO(NH_2_)_2_ + 6Ni(OOH) + H_2_O → N_2_ + 6Ni(OH)_2_ + CO_2_ + 6e^-^(2)

In light of the CV studies, it is evident that bitter gourd phytochemicals are highly effective for surface modification of nanostructured materials for a specific application. Consequently, bitter gourd could be used for surface modification of other materials for catalytic applications. In addition to these preliminary electrochemical tests, we have also performed comprehensive measurements of the non-enzymatic urea sensor in terms of kinetics, linear range, repeatability, reproducibility, selectivity, charge transfer kinetics, and real sample analysis using a variety of electrochemical methods.

For the study of diffusion-controlled behavior of modified electrodes, a variety of scan rates ranging from 10–390 mV/s were used to assess the behavior of sample 1 of NiO electrodes during the electrochemical process in 0.1 mM urea. An illustration of this can be found in Figure 3a. Increasing sweep scan rates produced successive linear peak currents for oxidation and reduction, as shown in Figure 3a. As can be seen in Figure 3b, a linear fit was made by plotting anodic and cathodic peak currents against the square root of the scan rate [19].

Furthermore, we have added a table describing the peak current ratio and peak separation potential for better understanding as given in Table 1. This verifies that the scan rate analysis describes the oxidation/reduction of urea involving the semi-reversible mechanism [59]. This is due to the fact weak urea adsorption onto the NiO nanostructures before it goes to the oxidation process on the process of NiO nanostructures produced with bitter gourd.

The working range of the presented non-enzymatic urea sensor was determined by measuring CV curves at a scan rate of 50 mV/s for various urea concentrations as shown in Figure 4a. As the concentration of urea increased in the electrochemical cell containing 0.1 M NaOH, the oxidation peak current of the CV curve was enhanced. Nevertheless, the proposed NiO nanostructures did not affect the reduction peak of the CV curve, suggesting that they were only oxidizing urea in alkaline environments. Moreover, the increase in urea concentration showed the shift in oxidation peak current to more positive potential increment and it may be connected to the electrode fouling at higher urea concentration. As shown in Figure 4b, a linear plot was produced by extracting the oxidation peak current against the corresponding concentrations of urea. The linear range of 1–9 mM for NiO’s urea sensor, along with a regression coefficient of 0.99, confirms NiO’s excellent analytical capabilities in developing non-enzymatic methods for the determination of urea.

Under the given equations [60,61,62], the limit of detection (LOD) and the limit of quantification (LOQ) were calculated.
LOD = 3 S/M(3)
LOQ = 10 S/M(4)

Herein, M shows the slope and S is the standard deviation. The measured LOD and LOQ were about 0.02 mM and 0.09 mM, respectively. The results presented in Figure 4 were repeated three times, hence the error bars are provided in Figure 4b. The obtained non-enzymatic urea sensor results were compared with the recently reported enzyme-activated and non-enzymatic urea biosensors, as shown in Table 2. According to Table 2, the previous studies mainly employed chemical routes for the synthesis, involving complicated steps and a large number of hazardous chemicals. While the presented nanostructured NiO material is obtained by the green chemical route with the use of minimum chemicals. This suggests low-cost fabrication, eco-friendly, environment-friendly, and scale-up synthesis. Another point about the published works on the enzymatic method is limited by the high cost of urease enzyme, multistep immobilization, and denaturation of urease enzyme. This reduces the storage life of urea-based biosensors. Based on this study, it is evident that the presented non-enzymatic urea sensor is highly sensitive, exhibits a broad linear range compared to non-enzymatic urea sensors, has a low limit of detection, and is inexpensive to manufacture. According to the comparative study, the previously reported non-enzymatic sensor [63] has a wide linear range, but it lacks the required analytical features. This can be seen from the fact that R^2^ = 0.96, which suggests doubts about the performance of the sensor. Our case, however, demonstrates excellent analytical features and well-described performance. Additionally, the presented urea sensor offered real-time urea quantification from a variety of real samples. It is evident that NiO nanostructures are economical, facile, and environmentally friendly with a linear range of 1.0 mM to 9 mM and a low limit of detection of 0.02 mM. Therefore, the proposed method is a potential and alternative method for quantifying urea in real biological fluids. We have measured the chronoamperometric response for various urea concentrations ranging from 0.1 mM to 5 mM and have observed that the current increases linearly with increasing urea concentration, as shown in Figure 5a. An excellent linear plot was obtained with the measured chronoamperometric current of each response time curve at 250 sec against urea concentrations ranging from 0.1 mM to 5 mM as shown in Figure 5b. Furthermore, we have used the chronoamperometric response of NaOH without adding urea and it was possible to conclude that the proposed configuration is capable of detecting urea very sensitively at 0.1 mM urea concentrations. Since amperometry is an electrochemical mode that is highly sensitive, it has supported the linear range in the CV mode; therefore, NiO nanostructures (sample 1) have a high potential to be investigated and tested for the analysis of real samples of urea. Additionally, the change in CV oxidation peak current was examined by adding the common interfering species, and the measured response is shown in Figure 5c. Clearly, the sequential addition of interfering agents did not result in any substantial change in peak current as shown in the inset of Figure 5c, confirming that sample 1 is extremely selective and sensitive only to urea in the presence of the same concentration of interfering agents. From zoom in view of the inset of CV curves shown in Figure 5c for each interfering suggests that the change in peak current is very negligible after the addition of interfering species compare to the peak current of urea. Hence high selectivity by sample 1 towards urea detection was demonstrated. NiO nanostructures have been reported to be used in many sensor applications, however, we have tested our material for many of the competing interferences of interests. A non-enzymatic sensor’s selectivity is primarily determined by its surface properties. The present study reports that bitter gourd extract was utilized to modify the surface of NiO material in order to achieve selective detection of urea. Furthermore, several CV cycles were measured for the same electrode in 0.1 mM urea concentration typically 25 cycles for the illustration of the excellent stability of the working electrode as shown in Figure 5d. There is negligible change in the peak current even after 25 cycles which could be assigned to the activation of the electrode.

Chronoamperometry was used to further investigate the stability of sample 1 in 0.1 mM urea during the time period of 300 s. The results indicated that NiO nanostructures displayed a high degree of compatibility with GCE. It is because they were able to detect the presence of urea without any fluctuation in current for the time period of 300 s, as shown in Figure 6a. The presented urea sensor exhibits a wide linear range as demonstrated by electrochemical impedance spectroscopy (EIS) conducted in 0.1 mM urea with bare GCE and pristine NiO. Figure 6b presents the Nyquist plots of NiO nanostructures assisted by bitter gourds in 10 and 15 mL. As a result of the experimental conditions, the frequency was swept from 100 kHz to 0.1 Hz, the sinusoidal potential was 10 mV, and the biasing potential was 0.5 V in 0.1 M NaOH. According to Figure 5b, the EIS data were fitted by equivalent circuits with well-defined circuit elements such as solution resistance (R1), charge transfer resistance (R2), and constant phase element (CPE). When an electrocatalytic interface is established between two phases, the charge transfer resistance between the two phases must be considered as a parameter to support the electrochemical reaction. The CPE is used to illustrate the behavior of the double layer, indicating that the fitted equivalent circuit of EIS data contains an imperfect capacitor. According to the Nyquist plot arc, charge transfer is relatively easy between electrodes during electrochemical reactions. We determined that the charge transfer resistance (R_ct_) for bare GCE, pristine NiO, and 10 and 15 mL of bitter gourd-assisted NiO nanostructure was 650 kΩ, 78 Ω, and 324 Ω, respectively. Due to its high conductivity and favorable charge transfer during urea oxidation, sample 1 displayed low charge transfer resistance and demonstrated rapid kinetics.

We also examined the repeatability, reproducibility, and stability of sample 1 in 1 mM urea solution by means of cyclic voltammetry at a scan rate of 50 mV/s. The repeatability was studied for the concentration range of 1 mM to 9 mM using sample 1 for three alternative days as given in Table 3. It can be seen that the non-enzymatic urea sensor demonstrated high capability to measure urea several times without losing linear range and LOD.

In order to obtain a deeper understanding of the performance evaluation, the electrochemically active surface areas of each sample were also evaluated with curves in the non-Faradic region at various scan rates as displayed in Figure 7a–c. In order to accomplish this, three samples were used, namely pure NiO, sample 1, and sample 2, prepared from bitter gourd peel extract in 0.1 mM urea. The electrochemical active surface area was estimated by dividing the difference in current density between the anodic and cathodic sides by 2. Then, we plotted that against the scan rate, the slope of which corresponds to the ESCA values of each sample as shown in Figure 7d. Clearly, sample 1 revealed a significant active surface area, which is probably responsible for its enhanced performance along with its low charge transfer resistance.

The practical aspects of the non-enzymatic urea sensor were studied using various real-life samples, such as blood, urine, dairy milk, curd, and wheat leaves. The results are presented in Table 4, Table 5, Table 6 and Table 7. It has been demonstrated that urea can be quantified in a wide range of actual samples, including clinical, dairy, and agricultural samples. This was completed using a NiO nanostructure prepared from 10 mL of bitter gourd juice. Moreover, excellent recovery rates of approximately 100% were observed in each sample, indicating the effectiveness of the non-enzymatic urea sensor when applied in real time.

## 4. Conclusions

To summarize, we have used green chemistry to synthesize NiO nanostructures using bitter gourd peel extract by low-temperature aqueous chemical growth method. An investigation of NiO nanostructures revealed the presence of a thin, flake-shaped cubic phase using bitter gourd extract. The XRD study has revealed a cubic phase and high purity of prepared NiO samples with bitter gourd extract. The electrochemical properties of different NiO samples prepared with 10 mL and 15 mL of bitter extract and pure NiO were studied for the non-enzymatic quantification of urea in the alkaline 0.1 M NaOH aqueous solution. We observed that the NiO sample 1 prepared with 10 mL of bitter gourd extract was found highly active for the oxidation of urea. The non-enzymatic urea sensor based on sample 1 was capable of detecting urea concentrations over a wide range of 1 to 9 mM with a detection limit of 0.02 mM when using CV mode. Chronoamperometric measurements have shown a linear range of 0.1 to 5 mM. The wide linear range and low limit of detection of urea on the surface of sample 1 were supported by the high amount of active surface area and the fast charge transfer rate. Our newly developed urea sensor has also demonstrated a high degree of sensitivity, selectivity, stability, and repeatability. The proposed non-enzymatic sensor configuration has been successfully applied to the determination of urea from various real samples, such as agricultural, clinical, dairy, and food samples and the obtained performance was found highly satisfactory. Compared with conventional methods for urea quantification, the fabricated sensor is low-cost, simple, highly sensitive, and selective. Biomass waste from bitter gourds could potentially be utilized for a new class of materials for tuning the catalytic properties and obtaining favorable surfaces for a wide range of applications such as energy conversion and storage systems.

## Data Availability

The data presented in this study are available on request from the corresponding author.

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
