# Peer review of "Green Synthesis of NiO Nanoflakes Using Bitter Gourd Peel, and Their Electrochemical Urea Sensing Application"

_micromachines, 2023, doi:10.3390/mi14030677_

Round 1

Reviewer 1 Report

Report on the manuscript micromachines-2300468-peer-review-v1 entitled “Green synthesis of NiO nanoflakes using bitter gourd peel, and their electrochemical urea sensing application”.

The submitted manuscript should be revised. The following points should be addressed:

1. The submitted manuscript should be revised to be free from editing or grammar errors.

2. The introduction should be supported by paragraph about recent literature studied NiO for urea sensing such as [RSC advances, 6(45), 39001-39006] & [Journal of Solid State Electrochemistry, 24, 3073-3081] & [Materials Letters 276, 2020, 128192] & [Materials Science and Engineering: B 240 (2019): 147-155].

3. Why the authors do hydrothermal treatment at 95°C and its below the boiling point of the used solvent?

4. More details in the experimental work should be supported such as the surface area of the applied working electrode and the amount of the electrocatalyst?

5. The JCPDS of NiO should be supported by the suitable reference such as International Journal of Hydrogen Energy

Volume 44, Issue 49, 2019, Pages 27001-27009.

6. The lattice parameters and strain should be estimated from XRD analysis.

7. What about the BET surface area of the prepared NiO nanostructures with bitter gourd.

8. There is two styles in the submitted manuscript after 3.2 paragraph. Please, fix the format style.

9. In figure 3A, the authors write that the applied scan rates from 10 mV/s and figure 3B started from lower than that value around 2.5 mV/s!!!

10. Figure 4 is not cyclic voltammetry may be its square wave or linear scan!

11. Figure 5B should be revised as only one curve clear and others were not clear and same for figure 6A.

12. For EIS analysis, the data before fitting and after fitting should be clear as reported in this reference: journal of alloys and compounds, 816, 2020, 152513.

13. Conclusion part should be clearer and focuses on the main achievement.

Author Response

Comments and Suggestions for Authors

Report on the manuscript micromachines-2300468-peer-review-v1 entitled “Green synthesis of NiO nanoflakes using bitter gourd peel, and their electrochemical urea sensing application”.

The submitted manuscript should be revised. The following points should be addressed:

We are thankful to the reviewer for useful comments and suggestions prior to publication

  1. The submitted manuscript should be revised to be free from editing or grammar errors.

Ans. The revised version of manuscript is well edited and proofread

  1. The introduction should be supported by paragraph about recent literature studied NiO for urea sensing such as [RSC advances, 6(45), 39001-39006] & [Journal of Solid State Electrochemistry, 24, 3073-3081] & [Materials Letters 276, 2020, 128192] & [Materials Science and Engineering: B 240 (2019): 147-155].

Ans. These citations have been added in the revised draft of manuscript

  1. Why the authors do hydrothermal treatment at 95°C and its below the boiling point of the used solvent?

Ans. The hydrothermal process at 95°C was carried in order to avoid the spoilage of solvent from the aluminum sheet covered beakers. This process was carried out in terms low temperature aqueous chemical growth and for better understanding,  we have replace hydrothermal process by low temperature aqueous chemical growth method.

  1. More details in the experimental work should be supported such as the surface area of the applied working electrode and the amount of the electrocatalyst?

Ans. The working of GCE was exhibiting an area of 3mm and it was used to deposit the catalytic material. A suspension of 10 µL with a loading mass of (0.2mg) was then applied onto 3mm of GCE and this information is added in the revised draft of manuscript

  1. The JCPDS of NiO should be supported by the suitable reference such as International Journal of Hydrogen Energy

Volume 44, Issue 49, 2019, Pages 27001-27009.

Ans. The recommended citation is properly cited with JCPDS card for the supporting used reference card.

  1. The lattice parameters and strain should be estimated from XRD analysis.

Ans. We tried to find the lattice parameters and strain, unfortunately we were not able to find some difference particularly in case of lattice parameters. We feel that they must be different but we are not yet clear about this calculation. However, it is a good comment and we try to seek more information on other XRD data analysis to calculate them and verify them by reported works.  But the point we tried to show in our paper was about the purity and phase of NiO obtained with bitter gourd, hence XRD data with standard JCPDS card and citation recommended by reviewer is presented in the revised draft of manuscript.

  1. What about the BET surface area of the prepared NiO nanostructures with bitter gourd.

Ans. We understand the reviewer comment about BET for the analysis of surface area of material prepared with bitter gourd, unfortunately we do not have access at the moment for this technique, hence we apologize  for this.

  1. There is two styles in the submitted manuscript after 3.2 paragraph. Please, fix the format style.

Ans. In the revised draft of manuscript uniform is made through the revised draft.

  1. In figure 3A, the authors write that the applied scan rates from 10 mV/s and figure 3B started from lower than that value around 2.5 mV/s!!!

Ans.  These corrections are made in the revised draft, we apologize for inconvenient during first submission.

  1. Figure 4 is not cyclic voltammetry may be its square wave or linear scan!

Ans. The results shown in Figure 4a are from CV curves because cathodic peak was also rising towards the anodic peak, hence making the presentation of CV data odd, therefore we took out the cathodic part of CV and only report the oxidation part of CV with rising concentration of urea.

  1. Figure 5B should be revised as only one curve clear and others were not clear and same for figure 6A.

Ans. In the revised draft of manuscript figure 5b is changed to 5c and zoo out view of other curves is shown in inset for the better understanding of obtained selectivity results. Also, we have provided additional data in the Figure 5 during the revised draft of manuscript.

  1. For EIS analysis, the data before fitting and after fitting should be clear as reported in this reference: journal of alloys and compounds, 816, 2020, 152513.

Ans. We thank you to reviewer for sharing nice reported work about EIS analysis.  When we see the reported EIS data and compare to our even fitted that has different behavior from the reported work. In our case we see significant charge transfer resistance for sample 2 and pristine and no proper semicircle behavior even after fitting. Whereas for the sample 1 has somehow small arch and after the fitted we were able to report the charge transfer resistance for three samples, consequently, the point of reviewer is interesting and we surely will adapt in our upcoming studies where we could have proper semicircle or two semicircle and of course EIS analysis is very complex phenomenon and it varies from sample to sample even for the similar material as in our case. However preparation of material under different conditions highly influence on the EIS behavior and even some times morphology of material has great impact on the charge transfer resistance. We hope reviewer would satisfy with our explanation.

  1. Conclusion part should be clearer and focuses on the main achievement.

Ans. In the revised draft conclusion is precisely presented.

Reviewer 2 Report

There are some main issue in manuscript that should be resolved precisely. A major revision is recommended according to following comments:

1)      Is it correct to name the synthesis procedure when only bitter gourd peel extract was used and whole procedure is the same as routine method? I mean, does only application of bio-waste materials in synthesis procedure make it green?

2)      Section 2.4, “In a subsequent step, 30 μl of 5% Nafion were poured into an ultrasonic bath and stirred for 20 minutes.” It should be corrected as “In a subsequent step, 30 μL of 5% Nafion were poured into NiO suspension in an ultrasonic bath and stirred for 20 minutes”.

3)      Section 2.3, please add details on measuring parameters: e.g. what is the resolution, for FE-SEM, how did sample deposition occur, what was the operating potential.

4)      Section 2.4, the concentration of KCl inside of reference electrode should be mentioned.

5)      Section 2.4, “It was conducted in this study to study electrochemical impedance spectroscopy (EIS)” should be corrected as “It was conducted to study electrochemical impedance spectroscopy (EIS)”.

6)      In case of Fig. 2, the CVs in both section “a” and “b” are almost the same. Their difference is very low. Although, the description of obtained data for sample 1 should be revised.

“It was estimated that the peak current for sample 1 in 0.1M NaOH was approximately 1.08 μA, whereas the peak current for sample 1 in 0.1M NaOH was approximately 1.27 μA. A difference of around 0.19 was observed between sample 1 in 0.1M NaOH and sample 1 in 0.mM urea, indicating sample 1 has significant catalytic properties for the oxidation of urea.”

Here, only 0.19 μA enhancement was observed. For concluding the accordance of electro-catalytic performance more evidence should be gained. It seems that the reduction peak in reverse scan is remained unchanged, that may be opposite of electrocatalytic effect.

7)      Page 7, what does “This is an area of high interest when it comes to surface and catalytic properties” mean?

8)      In case of scan rate effect, it has been mentioned that:

“Increasing sweep scan rates produced successive linear peak currents for the oxidation and reduction, as shown in Figure 3a. As can be seen in Figure 3b, a linear fit was made by plotting redox peak current against the square root of scan rate, resulting in a regression coefficient of 0.99. Therefore, the electrochemical kinetics was described via a surface controlled phenomenon [19]”

In Fig. 3b, the peak current vs. scan rate was plotted not square root of scan rate.

Did the peak potentials shift with scan rate increment?

9)      The heterogeneous rate constant, n, and diffusion coefficient of urea should be measured on all used electrodes.

10)  In case of Fig. 4a, why the peak potential shifted positively by increment of urea concentration?

11)  Page 8, second paragraph, “LLOD” should be corrected as “LOD”.

12)  In case of interference study, there are many report on development of sensor for ethanol, uric acid, glucose based on NiO nanostructures. How your synthesized modifier was able to selectively only determine urea? What feature has caused this selectivity?

13)  In Fig. 6, the current unit in vertical vector is “A”, is it correct? Also, the current value for sample 1are about a tenth as much as CVs of sample 2 and pure NiO. Check them again.

14)  In case of real sample analysis, Authors didn't give the evidence for the presence of urea in the real samples. Give the evidence for it. Only spiked samples were determined and the recovery efficiency of urea was presented.

15)  Is there any matrix effect determined in the urea detection in real samples environment using standard addition methods.

Author Response

Ms. No.: micromachines-2300468

Title: Green synthesis of NiO nanoflakes using bitter gourd peel, and their electrochemical urea sensing application

There are some main issue in manuscript that should be resolved precisely. A major revision is recommended according to following comments:

We are thankful to the reviewer for useful comments and suggestions prior to publication

  • Is it correct to name the synthesis procedure when only bitter gourd peel extract was used and whole procedure is the same as routine method? I mean, does only application of bio-waste materials in synthesis procedure make it green?

Ans. We thank the reviewer for his point. We have modified the word green throughout the text of manuscript. However, we are aiming to enhance the functional properties of material by using bitter gourd mainly and in this context we were calling it a green method for enhancing electrochemical properties of material.

  • Section 2.4, “In a subsequent step, 30 μl of 5% Nafion were poured into an ultrasonic bath and stirred for 20 minutes.” It should be corrected as “In a subsequent step, 30 μL of 5% Nafion were poured into NiO suspension in an ultrasonic bath and stirred for 20 minutes”.

Ans, This has been corrected in  the revised version of manuscript

  • Section 2.3, please add details on measuring parameters: e.g. what is the resolution, for FE-SEM, how did sample deposition occur, what was the operating potential.

Ans. Authors would like to thank the reviewer for the tip. We investigated the shape and orientation of nanostructured NiO, by depositing a few amount of  NiO sample on  the conducting carbon tape, using low resolution scanning electron microscopy at an accelerating potential of 20 kV.

  • Section 2.4, the concentration of KCl inside of reference electrode should be mentioned.

Ans. In the revised manuscript and the concentration of KCl of 3.0 Ml was added.

  • Section 2.4, “It was conducted in this study to study electrochemical impedance spectroscopy (EIS)” should be corrected as “It was conducted to study electrochemical impedance spectroscopy (EIS)”.

Ans. The reviewer point of view is considered. We thank the reviewer for his point. In the revised manuscript, it has been modified

  • In case of Fig. 2, the CVs in both section “a” and “b” are almost the same. Their difference is very low. Although, the description of obtained data for sample 1 should be revised.

“It was estimated that the peak current for sample 1 in 0.1M NaOH was approximately 1.08 μA, whereas the peak current for sample 1 in 0.1M NaOH was approximately 1.27 μA. A difference of around 0.19 was observed between sample 1 in 0.1M NaOH and sample 1 in 0.mM urea, indicating sample 1 has significant catalytic properties for the oxidation of urea.” Here, only 0.19 μA enhancement was observed. For concluding the accordance of electro-catalytic performance more evidence should be gained. It seems that the reduction peak in reverse scan is remained unchanged, that may be opposite of electrocatalytic effect.

Ans.  Figure 2 is explained and described more acurately in the revised draftFigure 2a shows CV curves for the different materials in the presence of electrolyte of urea for understanding the electrochemical performance of each material. The electrochemical activities of different materials in electrolytic solution show the obvious redox behavior. Furthermore, tt was estimated that the peak current for sample 1 in 0.1M NaOH was approximately 1.08 µA, whereas the peak current for sample 1 in 0.1M urea was approximately 1.27 µA. A difference of around 0.19 µA was observed between sample 1 in 0.1M NaOH and sample 1 in the absence of urea, indicating sample 1 has significant catalytic properties for the oxidation of urea. Furthermore, the CV curves of different material in the presence of 0.1mM urea are shown in Figure 2b, the enhancement in peak current for the sample 1 in the presence of urea is relatively higher compare to the pristine sample, indicating the favorable urea oxidation on the surface of sample 1. The reverse scan is not remaining unchanged for each material because in Figure 2, energy material has different scan rate peak current, suggesting that each material have certain electrocatalytic behavior which is further supported by the new tabled added by taking peak current ratio and peak separation potential ratio as shown in Table 1.

  • Page 7, what does “This is an area of high interest when it comes to surface and catalytic properties” mean?

Ans. Authors verification has been clarified in the revised manuscript  for better understanding, this suggests that the bitter gourd could use for the surface modification of other materials for the catalytic applications.  

  • In case of scan rate effect, it has been mentioned that:

“Increasing sweep scan rates produced successive linear peak currents for the oxidation and reduction, as shown in Figure 3a. As can be seen in Figure 3b, a linear fit was made by plotting redox peak current against the square root of scan rate, resulting in a regression coefficient of 0.99. Therefore, the electrochemical kinetics was described via a surface controlled phenomenon [19]”

In Fig. 3b, the peak current vs. scan rate was plotted not square root of scan rate.

Ans. Scan rate draft is modified and new additions are made. For this purpose, we have added a table describing the peak current ratio and peak separation potential for better understanding. This verifies that the scan rate analysis describes the oxidation/reduction of urea involves the semi-reversible mechanism.  This is due to the fact weak urea adsorption  onto the NiO nanostructures before it goes to the oxidation process on the process of NiO nanostructures produced with bitter gourd.

Did the peak potentials shift with scan rate increment?

Ans. This part is updated in the revised draft of manuscript and the peak potential was shifting due to possible weak urea oxidation with sweeping scan rate.

  • The heterogeneous rate constant, n, and diffusion coefficient of urea should be measured on all used electrodes.

Ans. This is very interesting comment from the reviewer. We have tried to understand it very much but we are still not clear to report in the revised manuscript. However we have provided the additional data analysis by peak separation current ratio and peak separation potential ratio in the tabular form for the convincing to the reviewer. Of course, the reviewer has given us useful comments about the calculation of heterogeneous rate constant n and diffusion coefficient of urea, but we face difficulties in calculation and have at the same mixed mind which electrodes reviewer wants to elucidate with rest to its comment. Yes, we will surely reach to conclusions by taking help from a person who could be familiar about the advanced electrochemical applications to collaborate with us in this comment and part of electrochemical data analysis.

  •  
  • In case of Fig. 4a, why the peak potential shifted positively by increment of urea concentration?

Ans.  The increase in urea concentration showed the shift in oxidation peak current to more positive potential increment and it may be connected to the electrode fouling at higher urea concentration.

  • Page 8, second paragraph, “LLOD” should be corrected as “LOD”.

Ans. This has been corrected in the revised draft of manuscript.

  • In case of interference study, there are many report on development of sensor for ethanol, uric acid, glucose based on NiO nanostructures. How your synthesized modifier was able to selectively only determine urea? What feature has caused this selectivity?

Ans. We agree with the reviewer comment there are many sensors reported on the NiO nanostructures, however, we have tested our material for many of the competing interference as described by bthe reviewer. The non-enzymatic sensors selectivity is mainly governed by the surface properties of material. In this case, we have modified the surface of presented NiO material using bitter gourd extract which played a vital role towards the selective detection of urea as reported in the this work.  This description has been added in the revised draft of manuscript.

  • In Fig. 6, the current unit in vertical vector is “A”, is it correct? Also, the current value for sample 1are about a tenth as much as CVs of sample 2 and pure NiO. Check them again.

Ans. In the revised draft of manuscript, this has been corrected.

  • In case of real sample analysis, Authors didn't give the evidence for the presence of urea in the real samples. Give the evidence for it. Only spiked samples were determined and the recovery efficiency of urea was presented.

Ans, We used a recovery method to support on our claim that presented urea sensor has capability to sense urea from real samples. However, there are still spaces to test the real sample by our reported method. For example, the pretreatment of real samples require some additional instrument before we test them with our  sensor. And those techniques are not available at the moment our lab. Hence we just used a recovery to highlight the possibility of material to be tested for the real sample analysis.

  • Is there any matrix effect determined in the urea detection in real samples environment using standard addition methods.

Ans. Yes, it is true and it is reason we could not get signal of our sensor for urea in real sample because there are many pretreatment methods are required to analyze real samples. The real sample contains many other proteins, facts and or even sugars etc which is complex systems and hence in such environment. It is very challenging for non-enzymatic sensor to be responded for specific  analayte.

Reviewer 3 Report

see attached file.

Author Response

Review comments

We are thankful to the reviewer for useful comments and suggestions prior to publication

Dear authors,

The manuscript micromachines-2300468 entitled 'Green synthesis of NiO nanoflakes using bitter gourd peel, and

their electrochemical urea sensing application' presents the development of a very interesting sensor capable of detecting the presence of urea from a green synthesis using a waste biomass. However, I suggest some modifications:

  • In the introduction, in my opinion, more emphasis should be placed on the use of biomass. This in my opinion makes a lot of difference to the quality of the work. Therefore, I suggest expanding the introduction a little bit by including the importance of the use of these compounds, pointing out the increasing attention in the use of these materials, and also quoting examples from the literature such as: 1039/D2NJ05582A, 10.1016/j.jelechem.2022.117071, Etc.

Ans. Authors would like to thanks the reviewer for his comments. In the revised draft we have highlighted the importance of biomass and the recommended citations  have added.

  • Format the text well (e.g. page 6).

Ans. Format of text is corrected in the page 6

  • Page 7 states 'As can be seen in Figure 3b, a linear fit was made by plotting redox peak current against the square root of scan rate, resulting in a regression coefficient of 0.99. Therefore, the electrochemical kinetics was described via a surface controlled phenomenon' but Figure 3b shows the exact values but it is not specified in the x-axis that it is the square root and furthermore the text states that it is controlled by a surface phenomenon when instead the linearity at the square root of scan rate indicates a diffusion phenomenon. This needs to be

Ans. During the revised draft and it has been corrected

  • the sensor has been tested in the range of 1 to 9 mM but what is the response at lower concentrations?

Ans. We have tested the sensor upto 0.1mM as can be seen from the chronoamperometry but we could not see  significant change in CV curves after 1 mM due to low sensitivity of CV compare the chronoametrometry. 

  • Figure 5a you can only see the analysis for sodium ion all the others cannot be seen. We need to improve the figure.

Ans. For better understanding the figure 5a has been modified and zoom out of peak signal of each interfering agent is shown in interference data measured through CV curves.  

  • For the chronoamperometry analysis, didn't you develop a calibration line?

Ans. In the revised draft the linear for chronoamperometry is provided

  • Also in figure 7 you can only see one line referring to urea the others cannot be seen in order to better understand the

Ans. In this figure 6a was used to illustrate the stability of electrode  only in the presence of 0.1mm urea

8) Have you carried out sensor characterizations by EIS and CV analysis in ferro/ferricyanide?

And. We thank the reviewer for the suggestion, however we did not study the EIS and CV

in ferro/ferriccyanide and we only anazlyed the sensor in NaOH electrolyte.

  • Have you examined the repeatability of the sensor on different days but tested the stability of the sensor to respond to the same concentration of urea for several consecutive cycles?

Ans. In the revised draft of manuscript we have already provided the repeatability of sensor performance  in Table 3 and stability for several consecutive cycles measured added in the revised draft of manuscript.  

  1. Reproducibility by developing multiple sensors (at least 3) to test the same urea concentration with multiple devices was investigated.

Ans. We have been working on the non-enzymatic sensor last several years and we noticed that reproducibility of sensor is mainly governed by the proper fabrication/modification of electrode then sensor is always reproducible. However, in this study we did not studied the reproducibility.

  1. Diagram 1 the structures refer to precise compounds while the name is generic it would be good to fix this. Numbers are also present and should be removed.

Ans. In the revised draft of manuscript, for better understand the general  information  about nature chemical compounds is described in the text of introduction and they have been defined in terms of stabilizing agents, capping agents, reducing agents which can modify the surface of nanostructured, consequently an enhanced electrocatalytic properties could be achieved. In the revised draft could be with many figures. Hence, for maintaining the number figures in paper the scheme 1 is removed.

Round 2

Reviewer 1 Report

Accept

Reviewer 2 Report

The revised manuscript can be accepted for publication in presenet format.